# Multifaceted Assessment of Porous Silica Nanocomposites: Unraveling Physical, Structural, and Biological Transformations Induced by Microwave Field Modification

**DOI:** 10.3390/nano14040337

**Published:** 2024-02-08

**Authors:** Aleksandra Strach, Mateusz Dulski, Daniel Wasilkowski, Krzysztof Matus, Karolina Dudek, Jacek Podwórny, Patrycja Rawicka, Vladlens Grebnevs, Natalia Waloszczyk, Anna Nowak, Paulina Poloczek, Sylwia Golba

**Affiliations:** 1Doctoral School, University of Silesia, Bankowa 14, 40-032 Katowice, Poland; 2Institute of Materials Engineering, University of Silesia, 75 Pulku Piechoty 1A, 41-500 Chorzow, Polandsylwia.golba@us.edu.pl (S.G.); 3Institute of Biology, Biotechnology, and Environmental Protection, Faculty of Natural Sciences, University of Silesia, Jagiellonska 28, 40-032 Katowice, Poland; daniel.wasilkowski@us.edu.pl (D.W.); anna.m.nowak@us.edu.pl (A.N.); 4Materials Research Laboratory, Silesian University of Technology, Konarskiego 18A, 44-100 Gliwice, Poland; krzysztof.matus@polsl.pl; 5Łukasiewicz Research Network, Institute of Ceramics and Building Materials, Cementowa 8, 31-938 Cracow, Poland; karolina.dudek@icimb.lukasiewicz.gov.pl (K.D.); jacek.podworny@icimb.lukasiewicz.gov.pl (J.P.); 6A. Chełkowski Institute of Physics, University of Silesia, 75 Pulku Piechoty 1, 41-500 Chorzow, Poland; 7Faculty of Chemistry, University of Latvia, Jelgavas Street 1, LV-1004 Riga, Latvia; 8Faculty of Chemistry, Silesian University of Technology, B. Krzywoustego Street 6, 44-100 Gliwice, Poland; natalia.szulc@polsl.pl

**Keywords:** porous silica spheres, surface area, silver nanoparticles, core–shell system, silver–silver carbonate heterojunctions, envelope, microwave treatment, microbial activity

## Abstract

In response to the persistent challenge of heavy and noble metal environmental contamination, our research explores a new idea to capture silver through porous spherical silica nanostructures. The aim was realized using microwave radiation at varying power (P = 150 or 800 W) and exposure times (t = 60 or 150 s). It led to the development of a silica surface with enhanced metal-capture capacity. The microwave-assisted silica surface modification influences the notable changes within the carrier but also enforces the crystallization process of silver nanoparticles with different morphology, structure, and chemical composition. Microwave treatment can also stimulate the formation of core–shell bioactive Ag/Ag_2_CO_3_ heterojunctions. Due to the silver nanoparticles’ sphericity and silver carbonate’s presence, the modified nanocomposites exhibited heightened toxicity against common microorganisms, such as *E. coli* and *S. epidermidis*. Toxicological assessments, including minimum inhibitory concentration (MIC) and half-maximal inhibitory concentration (IC_50_) determinations, underscored the efficacy of the nanocomposites. This research represents a significant stride in addressing pollution challenges. It shows the potential of microwave-modified silicas in the fight against environmental contamination. Microwave engineering underscores a sophisticated approach to pollution remediation and emphasizes the pivotal role of nanotechnology in shaping sustainable solutions for environmental stewardship.

## 1. Introduction

Metals have played dual roles throughout history. Some are integral components of pharmaceutical drugs despite their potential toxicity, even at trace concentrations [1,2,3]. On the other hand, many hazardous elements with notable examples including arsenic, lead, mercury, or noble metals like silver and copper, permeate the environment through various human activities associated with mining, industries, galvanization, metallurgy, automotive sector, fertilizers, etc. Here, broadly defined water and soil serve as reservoirs for these compounds [4]. Heavy metal contamination also diminishes soil fertility, with conventional remediation methods incurring substantial costs and often causing secondary environmental contamination [4,5]. Furthermore, an excess of environmental metals corresponds to increased bioaccumulation in living organisms and consequently affects the nervous, digestive, and reproductive systems [6].

Among the various heavy and noble metals that contribute to environmental pollution, silver stands out in laboratory and industrial applications due to its exceptional catalytic efficiency, surface activity, and active surface area [7]. Its properties make it widely utilized across diverse industries, including medical, cosmetic, apparel, food, construction, and household domains [8,9,10], and agriculture, where it exhibits excellent activity against fungal plant pathogens [11,12]. On the other hand, elevated concentrations of silver found in bottom sediments near municipal waste discharges and soils affected by diverse industrial activities [9,13] negatively impact plant growth in agricultural applications, as observed in Sunflower tissues (*Helianthus annuus* L.) [14]. In addition, the permeation of biological membranes through silver nanoparticles (AgNPs) and their accumulation in organs causes adverse effects such as inflammation in the lungs of rats [15]. Additionally, silver has detrimental impacts on osmoregulation in various organisms, e.g. zebrafish (*Danio rerio*) [16] and aquatic species like fish (*Oryzias latipes*), algae (*Raphidocelis subcapitat*), and plankton crustaceans (*Daphnia magna*) [13].

The complexities of silver toxicity constitute a multifaceted phenomenon influenced by many factors. Endogenous parameters, such as size, shape, chemical modifications, and exogenous factors (including dosage), play pivotal roles in the toxicological profile. Smaller silver nanoparticles exhibit heightened toxicity, potentially facilitated by their direct transgression of cellular membranes [17]. The evidence has been provided in studies of *Daphnia magna* [18]. Geometric intricacies, encompassing irregular shapes, surface charge, and Zeta potential, contribute to distinctive interactions with cellular entities, resulting in mechanical damage and overall toxicity [19]. Notably, the synthesis method plays a significant role, with green-synthesized AgNPs generally demonstrating lower cytotoxicity than their chemically synthesized counterparts [20]. The nature of silver usage is also dose-dependent. Low concentrations exhibit beneficial effects, while prolonged exposure to increasing concentrations escalates the toxicity of AgNPs [20]. Another critical factor impacting the toxicity of silver nanoparticles is their oxidation potential [21] and the subsequent release of Ag^+^ ions. These ions interact with proteins containing sulfur and nitrogen groups, leading to functional dysfunction [22]. The antibacterial mechanism of silver nanoparticles involves interactions with bacterial cell walls, thereby inhibiting biofilm proliferation [23,24]. Simultaneously, AgNPs induce DNA damage and impede cellular respiration by generating reactive oxygen species (ROS) [25]. It causes oxidative stress, oxidizing proteins and lipids, DNA damage, and ultimately leads to cell apoptosis. Furthermore, AgNPs can damage mitochondria through oxidative stress, directly affecting mitochondrial membrane proteins, impair cellular respiration, and decrease ATP production [26].

The amplification of these effects becomes more pronounced during the aggregation of silver nanoparticles. Hence, diverse strategies have mitigated this problem to protect silver against aggregation, including the core–shell structures such as Ag_2_O/Ag_2_CO_3_ [27], Ag_2_CO_3_@Al_2_O_3_ [28], Ag_2_S@Ag_2_CO_3_ [29], or Ag_2_CO_3_@Ti_2_O [30] composites. Ag@SiO_2_, Ag_2_O@SiO_2_, Ag_2_CO_3_@SiO_2_, or Ag_2_O/Ag_2_CO_3_@SiO_2_ composites are alternative and promising approaches. They are characterized by uniformly dispersed silver nanostructures within the silica carrier [31,32] and can be applied as the induced crystallization of Ag_2_O and Ag_2_CO_3_, yielding solid-state photocatalysts [33,34].

Following the Food and Drug Administration (FDA) regulations, silica is a safety material for food, biomedical, and as a carrier for drug delivery applications [35]. Its suitability stems from high chemical stability, modulatory capacity, and surface area augmentation through inherent porosity [36,37,38]. The capture potential of metal ions and the degree of their crystallization depend on pore size. Micro- (<2 nm) and mesoporous silica (2–50 nm) prioritize high surface areas, enhancing the metal-capturing potential [39,40]. Meanwhile, macroporous silica (>50 nm) offers a relatively lower specific surface area, influencing a lower metal-capturing potential [41,42]. Silica’s increased metal ion uptake potential is influenced by modification in its morphology [43,44]. One of the promising approaches is microwave heating. It can generate heat during the interaction of microwaves with dipole/ions in the reaction mixture [45].

In chemical synthesis, the microwave-assisted hydrothermal route involves microwaves interacting with matter. That creates a temperature gradient profile that penetrates and uniformly heats non-metallic materials due to their dielectric properties [46]. However, microwave energy is insufficient to break chemical bonds, and the corresponding electric field strength is too low to induce the organization of matter or a shift in chemical equilibrium [47]. The microwave-assisted hydrothermal route has also proven effective in reducing reaction temperatures or times, improving product yields, and enhancing material properties like phase, crystallinity, and particle size distribution compared to conventional methods [48,49,50,51].

Microwave radiation is also helpful for sintering oxide ceramics, non-oxide ceramics, metal powders, and intermetallics. It requires matter diffusion to fill pores between grains/particles [52,53]. Microwave radiation-assisted sintering is considered volumetric matter absorption at a very high heating rate, altering the activation energy of diffusion [54], reducing the sintering temperature, modifying phase transformation temperatures [55], controlling grain coarsening, inducing grain boundary melting, enhancing mass transport within grain boundaries, causing thermal stresses, and often resulting in the flattening of pores [56].

However, the precise effects of time or power of microwave radiation vary across different materials. Some studies have reported that longer treatment times, exceeding 30 min, led to increased complementary porosity [57], the formation of larger, interconnected, or more aggregated yet spherical particles [58,59,60], with a more crystalline structure and a lower specific surface area [61]. Conversely, shorter times with low-power radiation enabled the engineering of uniform silica particles [62] with the highest catalytic activity [61]. On the other hand, applying low power (P = 100 W) produced irregular particles with a rough surface. In comparison, higher power (P = 500 W) led to smaller, spherical, and more crystalline particles with a smooth surface and disordered structures [63].

Unfortunately, there is a fundamental lack of understanding regarding how microwave-field treatment, especially concerning time or power application, can manifest as non-thermal effects on the physicochemical features (such as size, shape, and porosity) of materials, especially silicas and the further sorption potential or crystallization of the metal nanoparticles. Therefore, the primary goal of this article is to comprehensively investigate the effects of microwave radiation-assisted sintering, including power and time, on the modification of the physicochemical and structural properties of silica carriers and to verify the sorption potential and direct crystallization of stable, bioactive Ag_2_O/Ag_2_CO_3_ heterojunctions enclosed within silica. This study extends to verifying the antimicrobial properties of the resultant material, utilizing Gram-positive and Gram-negative bacterial reference strains as a litmus test for its efficacy.

## 2. Materials and Methods

### 2.1. Estimation of the Particle Size and Zeta Potential

Commercially available porous silica nanoparticles (particle diameter: d_SiO2_ = 15–20 nm, pore size: d = 2–6 nm, pore volume = 0.8–1.5 cm^3^/g, purity: 99.5+%, surface area: 640 m^2^/g, density: 2.4 g/cm^3^) [64], suspended in water, underwent microwave field at various parameters (power: P = 150/800 W and time 60/150 s) (Table 1). Individual parameters taken from procedures described in the literature were modified [65,66]. Their selection considered maintaining stable global porosity and the microstructure of silica carriers [67]. Therefore, different microwave field durations at extreme power levels were chosen to evaluate the effect of field-assisted sintering technology on silica [52,53,67]. Moreover, the degrees of surface modification and overall morphology were selected, including specific surface area, pore size, and porosity.

The microwave-induced material aggregation facilitated the sonication of each silica carrier with an ultrasonic homogenizer (Omni Sonic Ruptor 400; PerkinElmer, Kennesaw, GA, USA) for 5, 10, 15, 20, and 25 min. Subsequently, the size of individually engineered silicas was determined using the Malvern Zetasizer Nano ZS particle size analyzer (Malvern Panalytical, Grovewood Road, Malvern, UK). Particle diameters were measured utilizing the dynamic light scattering (DLS) technique, employing a He-Ne laser with a wavelength of 633 nm, and calculated using the Stokes–Einstein equation (Equation (1)).

Measurements were executed in polystyrene cuvettes with a 1 cm optical path, and the detection angle was set at 173°. Each sample underwent five independent measurements, and the averaged values of the hydrodynamic diameter (d_H_) and polydispersity index (PDI) are summarized in Table 2. The assessments were conducted at various intervals (0, 3, 14, and 56 days) under consistent room temperature conditions to track potential aggregation. Samples of modified silica were suspended in ultrapure Millipore water. During the interim periods between measurements, the samples were securely stored at room temperature with controlled exposure to light. This approach ensured the accurate evaluation of silica diameters and aggregation behavior under the influence of the microwave field.
(1)$$d_{H}=kT/3\pi \eta D$$
where d_H_—hydrodynamic diameter, k—Boltzmann’s constant, T—absolute temperature, η—viscosity of the diluent, D—diffusion coefficient.

After the synthesis [32], the size of individually engineered silver–silica nanocomposites after prior sonication at 25 min with an ultrasonic homogenizer (Omni Sonic Ruptor 400; PerkinElmer, Kennesaw, GA, USA) was estimated using the Malvern Zetasizer Nano ZS particle size analyzer (Malvern Panalytical, Grovewood Road, Malvern, UK). Particle diameters were measured utilizing the dynamic light scattering (DLS) technique, employing a He-Ne laser with a wavelength of 633 nm, and calculated using the Stokes–Einstein equation (Equation (1)). The Zeta *ξ*-potential of silver–silica composite systems was measured on Malvern’s Zetasizer Nano ZS (Malvern Panalytical, Grovewood Road, Malvern, UK) at 25 °C and in a U-shaped cuvette (DTS1070). The detection angle was 173°. The electrophoretic mobility (UE) was determined using laser Doppler velocimetry (LDV), wherein the Zeta potential was recalculated using Henry’s equation (Equation (2)).
(2)$$U_{E}=2\varepsilon \xi f(K_{a})/3\eta$$
where U_E_—electrophoretic mobility, ε—dielectric constant, ξ—Zeta potential, f(K_α_)—Henry’s function, η—viscosity.

### 2.2. Sorption of Ag^+^ and Crystallization of Ag Nanoparticles

The UV-VIS spectra on individually prepared silicas, selected based on the DLS measurements, were meticulously recorded using a BioTek Synergy 4 multi-detection microplate reader (Agilent Technologies, 5301 Stevens Creek Blvd, Santa Clara, CA, USA). The experiments were conducted on Nunclon MicroWell Delta Surface 96-well plates (Thermo Fisher Scientific, 168 Third Avenue Waltham, MA, USA) over various time intervals.

For each experiment, 200 µL of aqueous silica solutions containing 1 mg of microwave-treated porous silica was used for blank measurements. In a separate experiment, a similar approach was followed with 200 µL of an aqueous solution with dissolved silver nitrate (purity above 99.9999% with trace metal basis; Merck Life Science Sp.z.o.o., Darmstadt, Germany) at a 5% concentration of Ag^+^. The experiment involved 200 µL of an aqueous solution with 1 mg of silica and at a 5% concentration of silver. Each experiment was conducted in triplicate using ultrapure Millipore water to eliminate the presence of undesirable ions. A calibration curve was executed using silver standard solutions spanning concentrations from 0.025 to 1 mg/L.

Absorption spectra were captured within the 250 to 450 nm wavelength range at 5 nm intervals. Measurements commenced at zero time and continued at 0, 0.5, 1, 2, 3, 4, 5, 6, 24, 26, 28, 30, 48, 52, 54, 72, 74, 78, and 168 h. Throughout this study, the plates were stored in a light-restricted environment at ambient temperature between measurements, ensuring the stability of the experimental conditions. Following each measurement, the background, represented by the spectrum of the pure solvent, was subtracted from the obtained spectra before analysis.

After the sorption experiment, sodium hydroxide (purity above 99.99%; Avantor Company, Gliwice, Poland) was added to an aqueous silver–silica up to pH = 10.4 to stimulate the crystallization of silver nanoparticles and enforce formation of Ag/Ag_2_CO_3_ heterojunctions (Table 1).

### 2.3. Determination of the Surface Area, Pore Volume, and Pore Diameter

The specific surface area (SSA_BET_), micropore volume (V_p_), area (S_P_), and average pore size (D_pore_) of silver–silica composite systems were determined utilizing a Gemini VII 2390a analyzer (Micromeritics Instruments Corp., Norcross, GA, USA). The measurements were meticulously conducted at the boiling point of nitrogen (−196 °C), and the final data were acquired through the application of the Brunauer–Emmet–Teller (BET) method and t-plot analysis. The inverse of the particle’s density (2.4 g/cm^3^ as delivered by the supplier [64]) was used to obtain particle volume. The total pore volume (TPV) was calculated from the gas sorption isotherm at p/p_0_ close to the saturation pressure (0.995 p/p_0_).

The porosity was computed as the TPV ratio to the sum of the TPV and solid particle volume (Table 3). Before measurements, samples underwent thermal treatment at 250 °C for three hours to eliminate gases and vapors adsorbed on the surface during synthesis. This degassing procedure was executed using a VacPrep 061 degassing system (Micromeritics Instruments Corp., Norcross, GA, USA). The degassed samples were not immediately analyzed but were kept at 60 °C. The instrument underwent verification before each use, involving the analysis of a carbon black reference material with a known surface area (P/N 004-16833-00 from Micromeritics, Norcross, GA, USA).

### 2.4. Structural Analysis

Diffraction data were meticulously acquired using the X’PertPro MPD PANalytical X-ray diffractometer, employing monochromatized CuK_α_ radiation (λ = 1.54 Å). The crystal structure refinement was executed through the Rietveld method using HighScore Plus software (version 5.1; Malvern PANalytical, Almelo, The Netherlands) and the ICCD PDF-4+ database. To facilitate quantification, a stabilized cubic ZrO_2_ served as an external standard. The Scherrer equation was applied to calculate the crystallite size and strain, with the broadening of the apparatus determined utilizing natural calcite possessing crystallites exceeding 10,000 Å. The crystallinity index was derived as the cumulative content of all crystalline phases.

The three-dimensional silica network quality and structural distortion within the silver carbonate were scrutinized by employing a WITec confocal Raman microscope CRM alpha 300 R (WITec Wissenschaftliche Instrumente und Technologie GmbH, Ulm, Germany), featuring an air-cooled solid-state laser (λ = 532 nm). The excitation laser radiation was directed into the microscope via a polarization-maintaining single-mode optical fiber with a 50 μm diameter, focusing monochromatic light onto the sample using an air Olympus MPLAN (100×/0.90 NA) objective. Raman-scattered light passed through a multi-mode fiber (50 μm diameter) into a monochromator with a 600 line/mm grating and a CCD camera, with the spectrometer monochromator calibrated using a silicon plate (520.7 cm^−1^). Raman spectra were accumulated at ten different locations, involving 20 scans with an integration time of 20 s and a resolution of 3 cm^−1^. Spectra with similar band arrangements were averaged, and post-processing analysis, including baseline correction and cosmic ray removal, was conducted using WITecProjectFive Plus (version 5.1.1; WITec Wissenschaftliche Instrumente und Technologie GmbH, Ulm, Germany). Peak fitting analysis was performed using GRAMS software (version 9.2; Thermo Fisher Scientific, Waltham, MA, USA).

### 2.5. Morphological Studies and Chemical Composition

STEM imaging, electron diffraction, and EDS analysis were meticulously captured with an acceleration voltage of 300 kV utilizing a Cs-corrected S/TEM Titan 80-300 FEI microscope outfitted with an EDAX EDS detector (FEI Company, Hillsboro, OR, USA). The high-angle annular dark field (HAADF) and bright-field (BF) detectors were used to perform high-resolution imaging at a few points, and local chemical content was achieved through the utilization of an energy dispersive spectrometer (EDS). Furthermore, a scanning electron microscope, enhanced with an energy dispersive X-ray spectrometer (SEM-EDS) (Phenom ProX; ThermoFisher, Eindhoven, The Netherlands), operating at an accelerating voltage of 15 kV, was employed to assess the chemical composition at the microscale.

### 2.6. Ag Ion Release

50 mg of Ag-SiO_2_ sample was immersed in 10 mL of deionized water (conductivity: 0.055 µS/cm) within 15 mL Eppendorf plastic tubes to estimate silver ion release kinetics. The immersion periods spanned 1, 3, 6, 12, 24, 48, and 72 h. Subsequently, 1.00 mL of the resulting solutions were carefully transferred into 50 mL Eppendorf plastic tubes and diluted to a total volume of 50 mL.

The silver ion concentration was employed using an Agilent 8900 inductively coupled plasma triple quadrupole mass spectrometer (ICP-MS QQQ) (5301 Stevens Creek Blvd, Santa Clara, CA, USA) with a micro-mist nebulizer and a helium collision cell. The instrument parameters were meticulously set: RF power—1550 W; sampling depth—8 mm; plasma gas flow rate—1.05 L/min; helium cell with gas flow rate—5 mL/min; nebulizer gas flow rate—0.90 L/min. A concentration calculation was performed through the calibration graph method with blank correction, utilizing five different silver standard solutions spanning concentrations ranging from 1.00 µg/L to 50 µg/L, all prepared from a stock standard solution (Merck, Darmstadt, Germany). The accuracy of the ICP-MS analysis was verified using two standard solutions (2.00 to 20.0 µg/L) that were sequentially measured after every ten samples, providing robust validation of the analytical process.

### 2.7. Microbial Studies

#### 2.7.1. Susceptibility to Antibacterial Agents

The toxicity of synthesized nanocomposites was evaluated in vitro against two model bacteria: Gram-negative *Escherichia coli* (ATCC^®^25922™) and Gram-positive *Staphylococcus epidermidis* (ATCC^®^12228™). The toxicological assessment encompassed two standard antibacterial study parameters: minimum inhibitory concentration (MIC) and half-maximal inhibitory concentration (IC_50_). The toxicity endpoint was defined as bacterial cells’ colony-forming ability on a solid growth medium when exposed to silver–silica nanocomposites.

Before employing the tested nanomaterials in biological assays, sonication of the aqueous suspension of each system in sterile Millipore Water was carried out using Vibra-Cell™ (5 min, 20 kHz) to prevent aggregation/agglomeration.

The toxicological experiment commenced with the inoculation of liquid LB medium (lysogeny broth medium; cat. 576832; Sigma-Aldrich, St. Louis, MO, USA) with bacterial culture obtained from the logarithmic growth phase, achieving a final OD_600_ equal to 0.1 (~10^7^ CFU m/L) in 12-well cell culture plates. Nanocomposites were individually added to the cultures, resulting in final concentrations ranging from 0 to 500 mg/L. The prepared samples were incubated for 24 h at 37 °C with continuous shaking at 140 rpm.

The antibacterial effect of the nanocomposites was quantified as the MIC value, representing the lowest concentration of nanocomposites inhibiting bacterial growth [68]. To determine the half-maximal inhibitory concentration (IC_50_), step-wise 10-fold dilutions in 0.85% NaCl from each microbial culture treated with specific systems were pipetted (30 µL) onto petri plates containing solid LB medium. After sub-culturing for 24 h at 37 °C, the colonies were counted, and the bacterial number in the nanostructure suspensions was calculated as colony-forming units per milliliter (CFU m/L). The entire experiment was replicated in triplicate, and the toxicological IC_50_ value was calculated using an online tool from https://www.aatbio.com (accessed on 22 December 2023) [69].

#### 2.7.2. Statistical Analysis

Data were statistically treated and presented as mean ± standard deviation values (mean ± SD). The statistical significance was followed using a one-way ANOVA and Tukey’s honest significant difference test (HSD). The substantial variations in experimental groups were represented in figures by annotated letters as *p* < 0.05 (*), *p* < 0.01 (**), and *p* < 0.001 (***) levels of significance. All experimental data were subjected to multivariate analysis. Hydrodynamic particle size was graphically presented using box-and-whiskers graphs. In turn, the multifaceted microbial analysis covering various aspects related to the features of silver nanoparticles and global silver–silica nanocomposites was performed using hierarchical cluster analysis (HCA) and summarized with the heat map. For this analysis, the data obtained were normalized according to Equation (3) [70]: (3)$$x_{\left(norm\right)}=\left(\left(x-x_{min}\right)\right)/\left(\left(x_{max}-x_{min}\right)\right)$$
where x_norm_—the value after normalization; x—the value before normalization; x_min_—the minimal value before normalization; x_max_—the maximal value before normalization.

All the graphical/statistical data were estimated using MS Office 2019 (Microsoft Inc., Redmond, WA, USA), OriginPro2023 (OriginLab Corporation, Northampton, MA, USA), and the Statistica 13.3 software package (TIBCO Software Inc., Palo Alto, CA, USA).

## 3. Results and Discussion

### 3.1. Microwave-Assisted Sintering: Deagglomeration, Specific Surface Area, Pore Diameter, Porosity

Due to preferential microwave absorption and material densification, silica nanoparticles underwent grain boundary melting and grain edge deformation following microwave-assisted sintering. Hence, before the experiment, each microwave-treated silica carrier underwent sonication to deagglomerate the system. Five sonication times were considered, and particle sizes were estimated using dynamic light scattering (DLS) measurements (Figure 1 and Figure 2; Table 2). Furthermore, the stability of the colloidal suspension with sonicated silicas was confirmed through DLS data, and the most stable systems underwent additional characterization to determine specific surface area, pore diameter, and porosity using gas porosimetry (Table 3).

DLS unveiled the average hydrodynamic diameters for all microwave-treated samples, ranging from approximately 500 nm (AR 2.2) to 650 nm (AR 1.1) after a short sonication period (t = 5 min). With an increase in sonication time, the average diameter consistently decreased, converging to a similar hydrodynamic value of about 360 nm for all samples at t = 25 min. Sample AR 2.2 had a slightly larger diameter of approximately 390 nm (Figure 1A; Table 2). The polydispersity index (PDI) values, indicative of particle size distribution, ranged from 0.7 after 5 min of sonication to 0.4 at 25 min (Appendix A). A detailed examination of diameters at short sonication times revealed significant differences, notably larger hydrodynamic diameters for silica carriers treated under a robust microwave field (P = 800 W). As sonication time increased, individual sample diameters approached each other, minimizing variability. No statistically significant differences between samples were observed at the longest sonication time, except for silica treated at P = 150 W and t = 150 s (AR 2.2) (Figure 1B).

Subsequently, the hydrodynamic diameter changes during storage of up to 56 days exhibited substantial dispersion for samples subjected to shorter sonication times, gradually diminishing with prolonged sonication (Figure 2). Interestingly, microwaved silicas, microwave-treated by t = 60 s, irrespective of power (AR 1.1 and AR 2.1), initially sonicated for 25 min, displayed remarkable stability with consistently similar diameter values during prolonged storage (Figure 2A,C). Conversely, the least stable silicas were those prepared using a microwave field with a power of P = 150 W and an irradiation time of t = 150 s (AR 2.2).

The DLS analysis indicates that a sonication duration of approximately t = 25 min proves highly effective in deagglomerating microwave-treated silica. These sonicated samples demonstrate enhanced stability during long-term storage, making them crucial for accurately determining specific surface area, pore diameter, porosity, and subsequent sorption experiments.

Gas porosimetry performed for silicas sonicated at t = 25 min revealed insightful patterns regarding the impact of microwave irradiation on silica morphology. A notable trend emerged, indicating that longer exposure times reduced the specific surface area, micropore volume, and micropore area (AR 1.2 and AR 2.2). In contrast, increased extended exposure times imply pore diameter and total pore volume, as evident in AR 1.1 and 2.1. Shorter exposure times (t = 60 s), regardless of the power (AR 1.1 and AR 2.1), prompted an augmentation in micropore volume (V_p_) and micropore area (S_p_) when compared to samples prepared at t = 150 s (AR 1.2 and AR 2.2) (Table 3).

The data translate into a silica porosity ranging from 30.97% to 57.59%, with lower values observed for carriers subjected to microwave radiation-assisted sintering at higher field power (AR 1.1 and AR 1.2). The silica densification showed sizes reminiscent of the producer-specified initial pore diameter (d = 2–6 nm), indicating the subsequent closure of the pore lumen. Conversely, an increase in irradiation time led to a rise in the global porosity of the material, perfectly correlating with the pore diameter (Table 3).

### 3.2. Sorption

Microwave-treated silicas, deagglomerated at an individually chosen time (t = 25 min), were assessed to verify each carrier’s sorption potential (Figure 3; Table 4). It was realized by preparing a mixture of silicas with Millipore water containing dissolved silver nitrate and kinetic UV-VIS studies. The concentration of silver ions accumulated within the carrier was determined after recalculating the band intensities (I_300nm_) based on the previously established calibration curve.

The time-dependent changes in UV-VIS absorption spectra facilitated the plotting of recalculated silver ion concentrations against time. These data were then fitted using a first-order kinetic equation (Equation (4)) for a comprehensive depiction (Figure 3):(4)y=y0+Ce−xt
where y_0_—total silver ion concentration (μg/L); C—silver ion concentration (μg/L); t—time of ion release (h).

Exponential fits prove highly effective in describing the experimental data for all investigated silicas, facilitating the determination of constant rates for the depletion of free silver ions in the solution or the accumulation of ions within the porous carrier. Detailed analysis reveals sorption capacities ranging from approximately 1 to 6 ppm over 40 to 70 h, contingent on microwave field power or time. Silicas exposed to higher microwave power (AR 1.1 and AR 1.2) exhibit twice the higher sorption capacity of those subjected to a lower microwave field (AR 2.1 and AR 2.2) (Table 4). Conversely, achieving higher capacitance necessitates a considerably longer time for complete pore saturation (Table 4).

### 3.3. Crystallization, Particle Size Diameter of Silver–Silica Nanocomposites

Following silica saturation with silver ions, each system underwent chemical modification by adding a sodium hydroxide solution, resulting in the crystallization of silver nanoparticles (Figure 3). Finally, four silver–silica nanocomposites after the sonication at a standardized time (t = 25 min) were subjected to DLS studies to estimate particle diameter and Zeta potential (Appendix A; Table 1).

DLS studies unveiled that the average hydrodynamic diameter of silver–silica nanocomposites falls within the range of 140 to 220 nm, contingent upon microwave field power and time (Appendix A; Table 1). Significantly lower values than for reference silicas, especially in AR 1.1 and AR 2.1, may arise from the susceptibility of the obtained silver–silica nanocomposites. Silicas treated at a higher microwave field (AR 1.1) disintegrated into individual components during sonification, leading to an underestimated average diameter value due to a second fraction related to free silver particles. Conversely, the impact of silver particle disintegration turned out to be less pronounced in other silver–silica nanocomposites, with a more robust bonding effect observed between silica pore walls and silver nanoparticles in microwave-treated silicas at a low microwave field and short time (AR 2.1) (Appendix A). Finally, the PDI values for all samples consistently fell below 0.4, indicating a monodisperse character with minimal aggregation in suspension (Zeta potential values ranging from −42 to −49 mV) (Appendix A).

### 3.4. Structural Analysis

The application of microwave-assisted sintering resulted in the development of four distinct nanocomposites with modified silica structures and crystallized silver nanoparticles. Consequently, two different techniques to examine the structural parameters were applied. Raman’s investigations yielded valuable insights into the carrier, delineating the three-dimensional silica network and its modifications under different microwave conditions (Figure 4A). In turn, the XRD data clarified the silver nanoparticles, focusing on crystal structure, lattice parameters, crystallite size, and strains associated with crystallization within the pore (Figure 4B,C; Table 5, Table 6 and Table 7).

Raman spectroscopy unveiled a weak band around 233 cm^−1^ in the low-frequency region for nanocomposites in which silica subjected to a longer time of microwave irradiation (AR 1.2 and AR 2.2) originated from silver oxide with a crystallographic structure closely related to Ag^I^Ag^III^O_2_ [71,72]. Conversely, the absence of low-frequency bands in the Raman spectrum of AR 1.1 and AR 2.1 precludes the identification of such oxidized forms of silver in these composites. In the low-frequency region (350–850 cm^−1^), bands associated with the Ag^I^Ag^III^O_2_ structure [71,72] overlapped with vibrations within ordered superstructures related to SiO_2_ basic silica units, such as the n-member ring structure with *n* > 4 [73,74] and vibrations within the Si_2_O_7_^6−^ unit [75]. The bands between 400 and 550 cm^−1^ correspond to mixed stretching and bending modes of Si–O–Si units, while those in the range 550–850 cm^−1^ arise from ring breathing modes [76]. Remarkably, these bands exhibited a significantly higher intensity in silicas subjected to a lower power of the microwave field and decreased as the power increased, suggesting a more pronounced impact of the microwave field on silica superstructures with low intermediate order.

The analysis of the high-frequency range (800–1250 cm^−1^) revealed bands describing silicate networks and the degree of depolymerization [77,78,79,80,81,82,83]. Notably, the bands around 905 and 977 cm^−1^, associated with Si-O* stretching modes in *Q*^1^ (Si_2_O_7_^6−^) and *Q*^2^ (Si_2_O_6_^4−^) units, were present in all Raman spectra. Additionally, bands around 805 and 1035 cm^−1^ related to *Q*^4^ units (SiO_2_) in fully polymerized silica structures were observed. Silicas that were subjected to a longer microwave treatment (AR 1.2 and AR 2.2) exhibited additional bands around 634, 686, and 1087 cm^−1^ originating from Si-O* vibrations within the metal-activated *Q*^1^ and *Q*^3^ units [79,84,85,86], indicating a potential capillary effect and enhanced metal capture.

Interestingly, silicas treated at a high microwave power and more prolonged exposure activated additional bands around 272 and 462 cm^−1^, stemming from translational modes within the silver carbonate structure [87,88,89,90]. Four bands at 687, 1063, 1350, and 1650 cm^−1^ were related to the ν_4_, ν_1_, and ν_3_ CO_3_^2−^ of the aragonite-type structure of silver carbonate [91]. Notably, the silver carbonate-related Raman bands for other samples were weak, practically overlapping with the silica network vibrations. This discrepancy suggests a potential core–shell structure formation [32,92], with a thicker carbonate shell for AR 1.1 and a smaller one for AR 1.2–AR 2.2 composites, supporting the molecular sorption potential of microwave-treated silicas (Figure 4A).

XRD measurements unveiled notable qualitative and quantitative variations in the phase composition of the samples, contingent upon the applied power and duration of the microwave field (Figure 4B). Quantitative analysis confirmed amorphous silica as the predominant phase in all tested materials, with SiO_2_ content estimated to exceed 95%. Simultaneously, the remaining composition included crystalline phases comprising two polymorphic forms of silver carbonate (Ag_2_CO_3_) and metallic silver nanoparticles (Table 5). No significant differences in the values of lattice parameters were observed when comparing individual phases (Table 6). The diffraction peaks of silver oxides were not identified, likely due to their low content or nanocrystalline nature.

In greater detail, one polymorph, β-Ag_2_CO_3_, exhibited a hexagonal crystal system (P3_1_c), while the other displayed a monoclinic structure (P2_1_/m), and their mutual content varied based on the applied microwave processing parameters. For instance, the amount of β-Ag_2_CO_3_ increased from 0.4% to 1.8% in the silver–silica nanocomposite subjected to more prolonged exposure to the microwave field at P = 150 W (AR 2.1 and AR 2.2). Conversely, a stronger microwave field showed practically no difference in this phase (AR 1.1 and AR 1.2). Meanwhile, the concentration of monoclinic Ag_2_CO_3_ decreased from 2.0% to 1.2% at a lower microwave field (AR 2.1 and AR 2.2) and increased from 1.5% to 2.1% at higher power (AR 1.1 and AR 1.2). Crystalline silver, with a cubic structure (Fm-3m), was consistently present at similar levels in all samples (AR 1.1–AR 2.2). Significantly, the crystallinity index, associated with the content of phases containing silver, declined as the power and time of the microwave field increased.

The analysis of crystallite size and strain for individual silver and carbonate phases showed a preferential growth of crystallites with values almost twice as high as those for other crystallographic directions of monoclinic and hexagonal silver carbonates, observed in [020] and [110], respectively (Table 7; Figure 4C). Moreover, carbonate crystallites were significantly larger than metallic silver (Table 7), indicating a directed growth of carbonates with highly elongated crystallites relative to the relatively small metallic silver crystallites. Conversely, more planar and elongated growth corresponded to lower strain than smaller and more compacted silver.

Another noteworthy observation was the lower crystallite dimensions of monoclinic silver carbonate as the power of the microwave field increased, in contrast to the hexagonal silver carbonates and metallic silver, which showed the opposite trend and lower dimensionality of crystallites at a lower microwave power. Moreover, notable deformation of the crystallites for monoclinic Ag_2_CO_3_ was observed, mainly due to an increase in strain and a reduction in their dimensions in the [020], [110], and [101] directions with more extended microwave treatment (AR 1.2 and AR 2.2). A similar trend was observed for crystalline silver in the [111] direction and for the hexagonal β-Ag2CO3 phase in the [110] direction. Conversely, changes towards [300] showed an increase in crystallite size with a longer exposure time of silica to the microwave field (AR 1.2 and AR 2.2).

The current research unveiled a consistent depolymerization of the silica network, irrespective of the specific microwave conditions applied (as observed through Raman spectroscopy). Simultaneously, it highlighted the modulating influence of microwave power and time on the silica surface, showing the selective crystallization of distinct silver carbonate polymorphs. Furthermore, it provided insights into crystallite size, directional elongation, and strain, as revealed by the XRD analysis. Interestingly, the crystallite size of silver carbonates exhibited elongation primarily in one direction for silicas subjected to shorter microwave sintering times, indicating a tendency toward pore flattening. Conversely, prolonged exposure times resulted in a reduction in the crystallite size of silver nanoparticles. Notably, microwave treatment demonstrated minimal impact on the quantity of metallic silver, marginal effects on crystallite size, and insignificant alterations in strain.

### 3.5. Microscopic Studies

The effects of microwave-assisted sintering and the crystallization of silver nanoparticles were comprehensively examined from macroscopic and microscopic perspectives, employing scanning electron microscopy (SEM) (Appendix A). Additionally, a more detailed investigation of silver nanoparticles was conducted using transmission electron microscopy (TEM) (Figure 5). Chemical analyses were conducted to assess silver content on a broader scale through SEM-EDS (Appendix A; Table 8) and locally through TEM-EDS (Figure 5 and Figure 6; Table 8).

The morphology of individually prepared Ag-SiO_2_ nanocomposites revealed a consistent pattern featuring irregularly shaped and variably sized objects (Appendix A). A more comprehensive analysis unveiled finely dispersed silver-related particles deposited onto the more agglomerated silica grains. The chemical mapping highlighted regions with the signal originating from silicon and oxygen, confirming the presence of the silica carrier without silver. Conversely, in the other areas, the signal corresponding solely to silver indicated the presence of metallic nanoparticles. The chemical analysis identified areas with calls originating from oxygen and silver, suggesting the formation of an oxidized form of silver. Further detailed analysis highlighted a higher diversity of samples subjected to high microwave power with the prevalence of areas containing oxidized silver over metallic (AR 1.1) or a coexistence of more elevated areas of metallic and well-dispersed smaller regions of oxide particles (AR 1.2). Samples prepared at lower power of microwave field (AR 2.1 and AR 2.2) showed a more proportional distribution of metallic silver intermixed with silver oxides.

The SEM-EDS analysis confirmed silica’s potential to uptake silver with a maximum capacity estimated below 3.0.% (Table 8). The highest silver content was evident in nanocomposites synthesized using low microwave irradiation power (AR 2.1 and AR 2.2). Conversely, samples exposed to higher microwave power (AR 1.1 and AR 1.2) exhibited lower silver content. Furthermore, the lowest metal concentration was found in the system exposed to high power and extended duration of the microwave field (P = 800 W, t = 150 s), measuring at 1.0 ± 0.1 at.%.

Consistent with the macroscopic analysis, the TEM micrographs unveiled the uniform distribution of Ag nanoparticles with varying morphologies in shape and size (Figure 5). The highest occurrence of aspherical silver nanoparticles was found in silicas subjected to more time-extended microwave irradiation at P = 150 W (AR 2.2). Conversely, the lowest degree of aspherical particles was observed in a sample exposed to a short microwave treatment at the higher power of AR 1.1. Extending the exposure time to the microwave field contributed to an increase in the percentage of aspherical nanoparticles.

Variations in microwave irradiation power and duration profoundly impacted the external and internal silica surfaces, delineating unique trajectories for crystallizing silver particles. Prolonged exposure to microwave radiation led to a notable increase in nanoparticles enveloped within the structure, culminating in core–shell configurations with a metallic silver core and a silver carbonate shell (Table 9). The envelopes demonstrated a parallel sensitivity by mirroring the behavior of microwave-sensitive crystallites. Silica initially exposed to lower microwave power fostered the formation of more rounded envelope structures, while samples subjected to higher microwave power manifested more square- or rectangular-shaped enveloped configurations (Figure 6). Prolonged irradiation resulted in the crystallization of more irregular particles with an increased propensity for aggregation, as observed in AR 1.2 and AR 2.2. Notably, square-shaped silver nanoparticles were identified in the case of nanocomposites treated with a microwave field at P = 800 W (Figure 6).

A detailed analysis, including the estimation of a diameter of spherical Ag nanoparticles (d_s_) and the surface area of spherical (S_s_), aspherical particles (S_as_), and enveloped structures (S_e_), was performed based on the individual distribution histograms (Figure 7A and Appendix A; Table 9). It is crucial to note that all systems exhibit significant diversification, and the discussed features do not apply to all particles. Concurrently, the mean diagonal, median area and standard errors were employed to assess the envelope structures (S_e_) (Table 10). The envelopes’ diagonal size was similar in AR 1.2, AR 2.1, and AR 2.2, except for AR 1.1. Intriguingly, the applied microwave features (P = 800 W, t = 60 s) led to the co-crystallization of a volumetric, highly structured envelope with a diagonal and envelope area (AR 1.1).

The average particle size of spherical silver particles ranged from d = 2.05 ± 0.03 nm (AR 1.1) to d = 10.19 ± 0.36 nm (AR 2.2) (Figure 7A; Table 9). Regardless of the power of the microwave field, an extension of the exposure time resulted in an increase in the average diameter of the silver nanoparticles by approximately 1–2 nm. Furthermore, silica subjected to high microwave-assisted sintering favored crystallization of low dimensional silver nanoparticles, compared to the silicas exposed to low microwave power (Figure 7).

The diffraction pattern shows cell parameters a_0_ = 4.085(7) Å and space group Fm-3m. Interplanar d-spacing pointed to the prevalence of locally occurring metallic silver in AR 2.1: Ag_(220)_ and AR 2.2: Ag_(111)_, as well as silver oxide for AR 1.1: AgO_(112)_ and AR 1.2: AgO_(011)_ (Figure 7B).

TEM-EDS showed lower metal content within the silicas microwave-irradiated at lower power and higher for nanocomposites synthesized at a high microwave irradiation power and t = 150 s (AR 1.2). Moreover, TEM-EDS showed that the silver content increased locally at longer silica times in the microwave field. It is observed at a higher power with almost twice the higher concentration (Table 8).

### 3.6. Kinetics of the Ag^+^ Release

To develop more applicable features and establish correlations with the engineered systems’ antibacterial properties, a kinetic analysis was conducted to examine the release of silver ions from the individually prepared nanocomposites, as confirmed by ICP-MS analysis (Figure 8).

The ICP-MS results unveiled a time-dependent kinetic pattern of silver ion release from the silver–silica nanocomposites into the water solution. The kinetic data revealed a two-step desorption process, with the behavior stemming probably from three distinct fractions of silver nanoparticles: (i) those bonded to the silica surface outside the pores; (ii) those crystallized within the nano-pores in the form of spherical or aspherical particles within an envelope; and (iii) those inside or outside surface-bonded in the form of variable-shape particles with and without an envelope. As a result, various kinetic models were thoroughly considered to determine the most appropriate fit for the experimental data [93,94,95]. Hence, an exponential function (Equation (4)) and the pseudo-first-order model (Equation (5)) were investigated as the most reliable to analyze the comparative time of silver release from the tested samples (Figure 8) [43,46,47,48].
(5)$$q\left(t\right)=q_{e}\left(1-e^{-k_{1}t}\right)$$
where q_e_—the number of ions after equilibrium of adsorption (µg/L); q(t)—the number of desorbed ions at time t (µg/L); k_1_—pseudo-first rate constant (1/h).

A comparative analysis of the two proposed models unveiled slight discrepancies in the values of silver ions released into the solution (Table 11). Both models exhibited a similar change trend, accounting for statistical uncertainty. However, the content of silver ion release from the composites decreased almost twofold when silica underwent a higher microwave field (AR 1.1 and AR 1.2), with a marginal impact on samples exposed to a weaker microwave field (AR 2.1 and AR 2.2). Given the high similarity in the observed trends, an exponential model was employed for further comparative analysis.

Upon closer examination, the content of silver ions released from AR 1.2, featuring the lowest values among all samples, stood out significantly. This observation aligned closely with the SEM-EDS data, which indicated the lowest atomic silver concentration in AR 1.2. Conversely, the nanocomposite treated with high microwave power but a short exposure time (AR 1.1) exhibited a distinct profile, showing the most extended release duration among all microwave-treated samples, totaling approximately 38.99 ± 5.20 h (Table 11). AR 1.1 demonstrated the highest concentration of released Ag ions at 1124.75 ± 66.03 µg/L and correlated to TEM micrographs, revealing smaller Ag nanoparticles than in the other nanocomposites. Notably, there is an increase in silver ion content in time for AR 1.1 (P = 800 W, t = 60 s), contrasting with the opposite trend for AR 1.2 (P = 800 W, t = 150 s). The observations for the number of released ions and the extension of the release time in the case of t = 150 s (AR 2.1) provided further insights into the nuanced effects on silver ion release.

According to the kinetic studies, the most promising nanocomposites for short-time experiments are those prepared under low-time microwave-assisted sintering conditions, exhibiting substantial silver ion release. Conversely, the silicas subjected initially to longer microwave irradiation durations demonstrated enhanced stability of the final nanocomposite over time, resulting in a sustained release of silver ions. Intriguingly, the highest content of silver ions released was observed in the systems treated at high power, emphasizing the influence of microwave power on the desorption kinetics.

### 3.7. Bacterial Viability Assay on Bacterial Strains Exposed to Various Nanocomposites Silver–Silica

The cytotoxicity investigation using minimal inhibitory concentration (MIC) and half-maximal inhibitory concentration (IC_50_) meticulously employed two model microorganisms—*E. coli* and *S. epidermidis* (Table 12).

MIC data revealed more excellent resistance of the *E. coli* strain to the nanocomposites than *S. epidermidis*. Specifically, AR 1.2, AR 2.1, and AR 2.2 samples exhibited the most pronounced inhibitory effects on cell growth at 150 mg/L concentrations. Conversely, *S. epidermidis* displayed heightened sensitivity to AR 2.2, inhibiting cell growth at a concentration of 80 mg/L. AR 1.1, AR 1.2, and AR 2.1 demonstrated similar effectiveness to *S. epidermidis* at a concentration of 120 mg/L. In turn, the IC_50_ for *E coli* unveiled the toxicity assessment in increasing order of toxicity: AR 1.2 > AR 2.1 > AR 2.2 > AR 1.1, while the following represents the antibacterial efficacy against *S. epidermidis* in increasing order of toxicity: AR 1.2 > AR 2.1 > AR 1.1 > AR 2.2. In summary, AR 1.1 and AR 2.2 emerged as nanocomposites with the highest effectiveness, exerting half their maximal inhibitory growth effect.

MIC and IC_50_ values indicated that the obtained nanocomposites exhibited superior antibacterial properties against *S. epidermidis* compared to *E. coli*. *S. epidermidis*, as a Gram-positive bacterium within the *Staphylococcus* genus, exhibits the capability of biofilm production with a robust cell wall comprising lipoteichoic acids and polysaccharides and tends to be more susceptible to the bactericidal properties of silver nanoparticles, as evidenced in various studies on Gram-positive and Gram-negative bacteria [96,97,98,99].

The cluster analysis provides insights into the intricate relationships between nanocomposites and the assessed parameters for *E. coli* and *S. epidermidis* bacterial strains (Figure 9 and Figure 10). In the dendrogram projection for *E. coli* (Figure 9), two distinct groups emerge, characterized by five and ten parameters, and the % cube stands out as the most differentiating variable in the first group. In contrast, pore surface (S_p_) and specific surface area (SSA_BET_), along with the microbial *Ec* IC_50_ parameter, were marked in the second group. Similarly, the cluster analysis for *S. epidermidis* identifies two primary groups, with the first strongly characterized by toxicological parameters S_e_ IC_50_ and S_e_ MIC and the second reflecting the profile observed for *E. coli* (Figure 10).

In the context of cluster analysis, physicochemical parameters significantly impacting MIC and IC_50_ values for individual bacterial strains were isolated. As a result, the *Ec* MIC value was strongly influenced by nanocomposite components related to the silver, involving shape, sphericity, and the amount of Ag ions released (Figure 9). The results suggest a likely influence of a direct mechanism related to the action of Ag ions and a direct one concerning nanoparticles [100]. Wang et al. demonstrated the toxic effect of Fe_3_O_4_@SiO_2_@Ag structures on *E. coli* via Ag^+^ ions [101]. Similar results were obtained for spherical SiO_2_@Ag material [97] and Ag@QHMS material [102].

Conversely, the MIC value of *S. epidermidis* appears dependent on parameters related to the system’s shape, sphericity, and silver content, indicating a direct effect of Ag nanoparticles on bacterial cells. Kang et al. demonstrated the high efficiency of intracellular AgNPs killing by endocytosis of small nanoparticles, up to 40 nm, by Gram-positive bacteria [103]. Swolana et al. found more significant toxicity against *S. epidermidis* for silver nanoparticles with a d = 10 nm in diameter [104]. On analyzing the physicochemical parameters associated with IC_50_ for both bacterial strains, silica nanocomposites, the size of AgNP, and silver content strongly affect *Ec* IC_50_. Hence, the synthesized nanocomposites impact the *Ec* IC_50_ through the mechanical effect related to the silica and silver nanoparticles. Similar observations highlighted by Gibala et al. illustrate the beneficial effect of stabilizing silver nanoparticles with additional agents on the bactericidal properties of the nanoparticles [105]. Gouyau et al. found the bactericidal activity of AgNPs against *E. coli* due to electrostatic interactions with cell wall lipopolysaccharides [106]. In turn, Morones et al. reported the effect of silver nanoparticles on Gram-negative bacteria. Their study made it possible to observe the interference of 1–10 nm AgNPs in the composition of the bacterial membrane, disrupting its function [107]. On the other hand, the properties of the silica and silver nanoparticles, including the amount of Ag^+^ ions released, influence the *Se* IC_50_ value, suggesting a synergistic effect of the silica, silver nanoparticles, and Ag^+^ ions.

Considering hierarchical cluster analysis for *E. coli*, AR 1.1 was distinctly separated, while AR 1.2, AR 2.1, and AR 2.2 form a thematically similar cluster (Figure 9). Conversely, a different pattern emerges for *S. epidermidis*, where cluster analysis groups AR 1.1, AR 1.2, and AR 2.1 into one cluster. At the same time, AR 2.2 was independently distinguished (Figure 10). The nanocomposite with silica initially microwave-treated at a high microwave field and short time (P = 800 W, t = 60 s) is characterized by the silver–silver carbonate core–shell system with the largest envelope area from all samples, exhibits the most potent antibacterial properties against *E. coli* among all tested nanocomposites and is the second most effective against *S. epidermidis*. The core–shell system with an enveloped Ag_2_CO_3_ in the AR 1.1 plays a crucial role against Gram-negative bacteria, enhancing the toxic effect through its photocatalytic properties when exposed to visible light [108,109]. This effect was also strongly correlated with the possibility of generating reactive oxygen species (ROS) that, combined with Ag ions released from the core–shell system, contribute to the oxidation processes of bacterial cell components, leading to cell death [110].

## 4. Conclusions

The application of a microwave field has profoundly impacted the silica carrier’s physicochemical, structural, and biological attributes. Utilizing DLS coupled with sonication at different times allowed the estimation of a sonication time of about t = 25 min for microwave-treated silicas, ensuring their stability in colloidal solutions over an extended period. Our studies showed that microwave radiation demonstrated a dual impact, reducing specific surface area while concurrently enlarging the pore diameter of the silica. Kinetic sorption studies illustrated that silicas exposed to higher microwave power (P = 800 W) exhibited twice the sorption capacity of those subjected to a lower microwave field (P = 150 W). An increase in the time of microwave field irradiation resulted in a rise in sorption capacitance. The optimal parameters were obtained at P = 800 W and t = 150 s.

Modifying the environment stimulated the crystallization of silver ions sorbed within the carriers towards forming silver nanoparticles. Our outcomes illustrated that the applied microwave features (P = 800 W, t = 60 s) led to the co-crystallization of a volumetric, highly structured envelope. The diagonal and envelope area confirmed the formation of an enveloped silver carbonate-related shell. Other structural analyses unveiled the formation of nanoparticles in three states: metallic and carbonates, with all systems displaying a stable Ag/Ag_2_CO_3_ heterojunction in a core–shell structure. Additionally, silica was exposed longer to microwave irradiation, stimulating silver oxide’s crystallization with a crystallographic structure closely related to Ag^I^Ag^III^O_2_.

Microscopic investigations unveiled that a lower microwave field power (P = 150 W) induced the crystallization of larger metallic silver nanoparticles, while prolonged irradiation times (t = 150 s) reduced the nanoparticle sphericity. Moreover, an extension of the exposure time increased the average diameter of the silver nanoparticles, considering the spherical Ag particle size. Notably, a smaller silver particle size indicates that metallic silver serves as the core of the crystallized system. Chemical investigations revealed a lower silver content within silicas microwave-irradiated at P = 150 W and a higher content for silicas subjected to microwave sintering at P = 800 W and t = 150 s.

According to kinetic studies, the most promising nanocomposites for short-time experiments exhibiting substantial silver ion release are those prepared under low-time microwave-assisted sintering conditions (P = 150 W). Conversely, the silicas subjected to more extended microwave irradiation (t = 150 s) demonstrated enhanced stability of the final nanocomposite over time, resulting in a sustained release of silver ions. Intriguingly, the highest content of silver ions released was observed in the systems treated at high power (P = 800 W), emphasizing the influence of microwave power on the desorption kinetics.

Toxicological assessments enabled the determination of MIC and IC_50_ for the tested nanocomposites against *E. coli* and *S. epidermidis* strains, correlating with the release of silver ions in the aqueous environment. Interestingly, the results highlighted an enhanced sensitivity of the *S. epidermidis* strain compared to *E. coli*. The most substantial toxic effects against *E. coli* were observed in the nanocomposite where the silica carrier was exposed to a high-power microwave field for a short time. Conversely, for *S. epidermidis*, the system in which the silica carrier was exposed to a lower-power microwave field for an extended period exhibited the highest toxicity.

Beyond its antimicrobial effects, our research underscores the efficacy of porous silica modified with a microwave field in capturing silver compounds from aqueous environments. This innovative approach opens potential applications in environmental remediation processes. Moreover, the formation of stable nanostructures within the silica holds promise for utilizing absorbed metal compounds endowed with bactericidal properties. This dual functionality positions our developed nanocomposites as versatile and impactful tools in environmental and biomedical applications.

## Figures and Tables

**Figure 1 nanomaterials-14-00337-f001:**
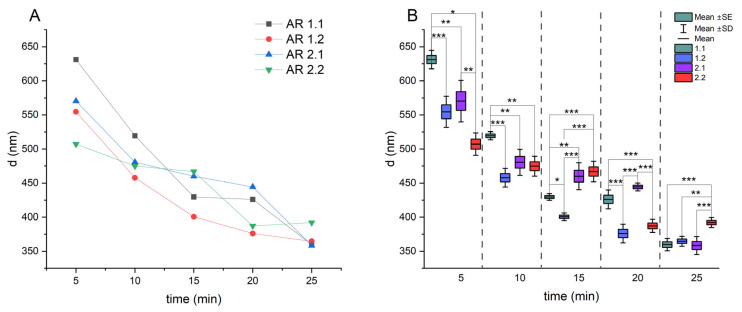
(**A**,**B**) Hydrodynamic averaged diameter (d_H_) of microwave-treated silicas depending on sonification time (t) without (**A**) and with (**B**) statistically significant differences (* *p* < 0.05; ** *p* < 0.01; *** *p* < 0.001). Data shown directly after sonication (t = 0 day).

**Figure 2 nanomaterials-14-00337-f002:**
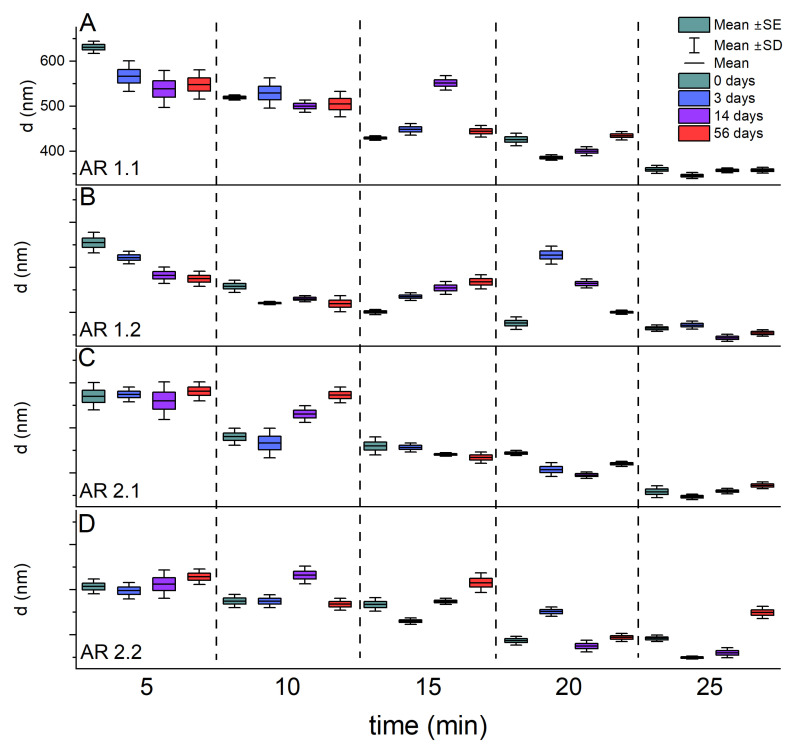
(**A**–**D**) Hydrodynamic averaged diameter (d_H_) dependence of sonification time (t) for (**A**) AR 1.1, (**B**) AR 1.2, (**C**) AR 2.1, (**D**) AR 2.2 during storage up to 56 days.

**Figure 3 nanomaterials-14-00337-f003:**
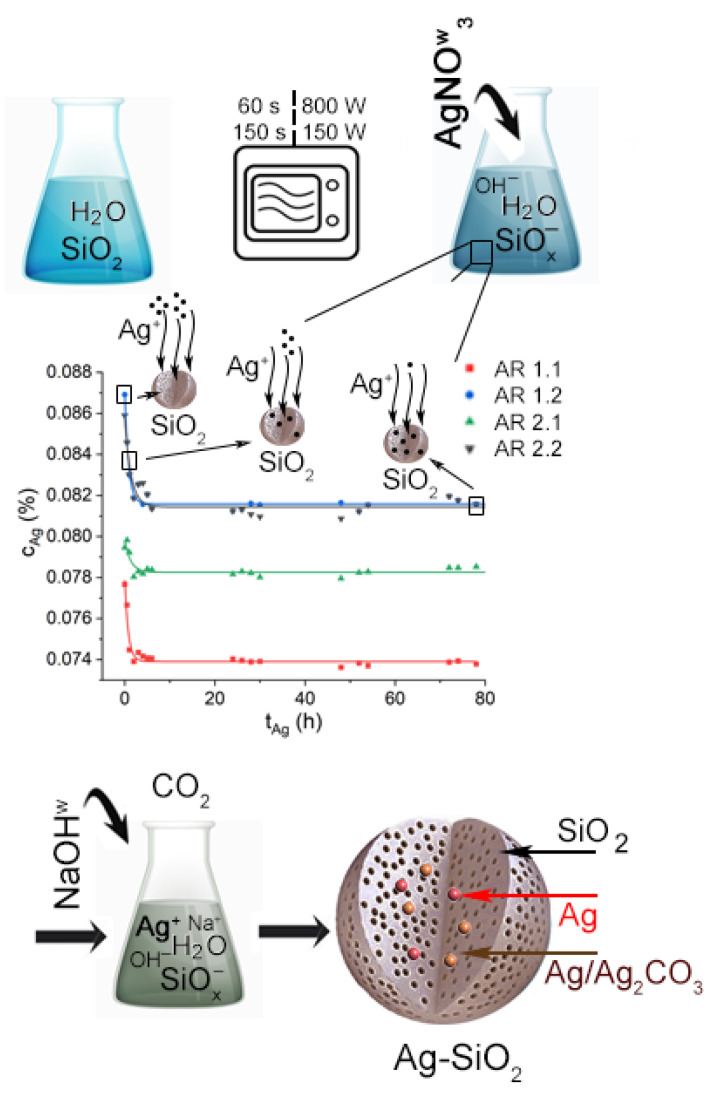
Schematic illustration of sorption experiment with kinetic curve summarized silver ion concentration (C_Ag_) sorbed by the microwave-treated silica in time (t_Ag_) and induced crystallization of silver nanoparticles within the silica pore walls.

**Figure 4 nanomaterials-14-00337-f004:**
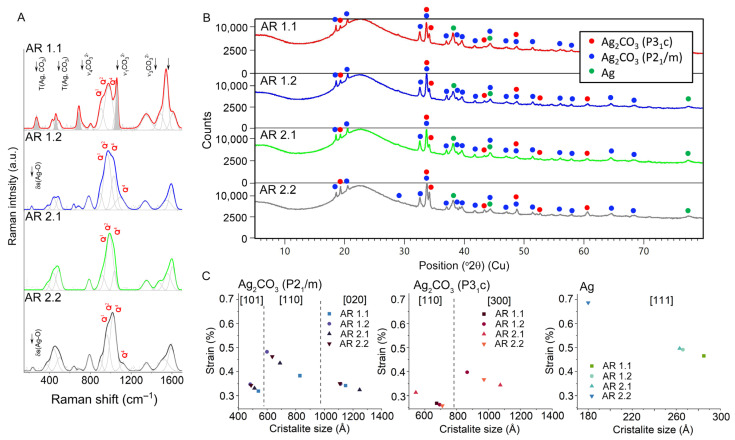
(**A**) Raman spectra, (**B**) XRD patterns, and (**C**) crystallite size and strain of the silver phases within the Ag-SiO_2_ nanocomposites. Individual colors on all figures correspond to the individual composites. Dash-line Raman spectra represent the silica, while the straight-line spectra represent the silver carbonate.

**Figure 5 nanomaterials-14-00337-f005:**
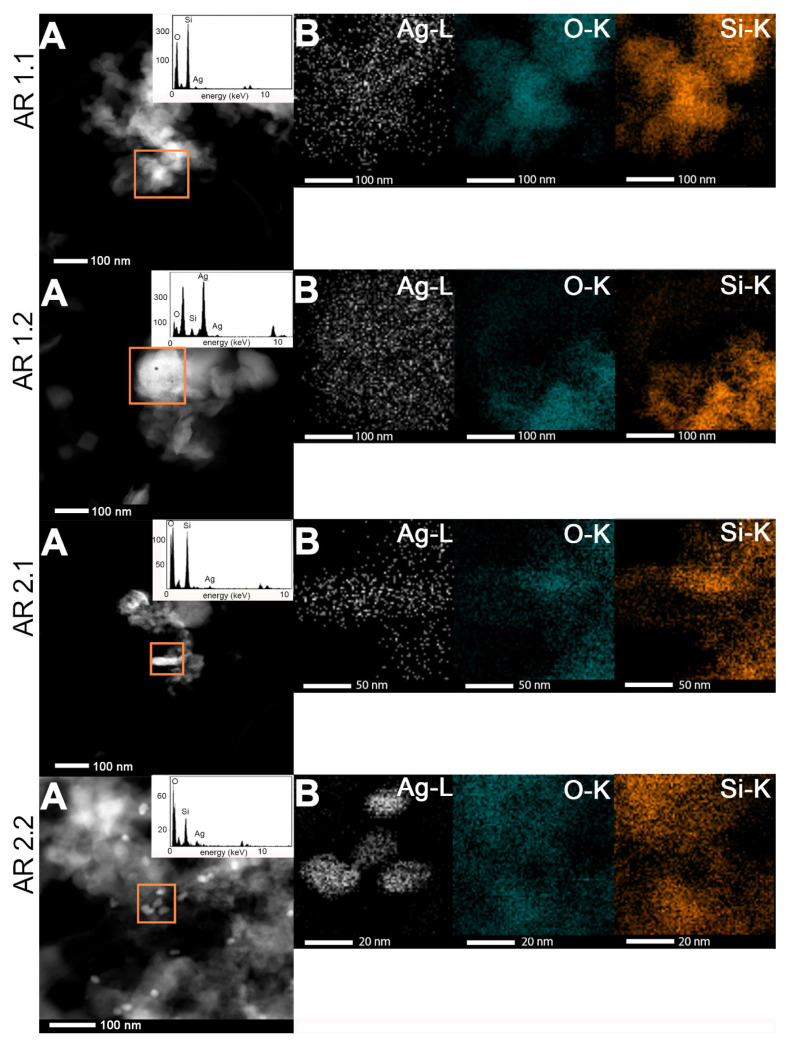
(**A**) HRTEM micro-images with TEM-EDS spectrum in the inset and (**B**) HRTEM chemical mapping patterns of the nanocomposites gathered from the region marked by orange boxes.

**Figure 6 nanomaterials-14-00337-f006:**
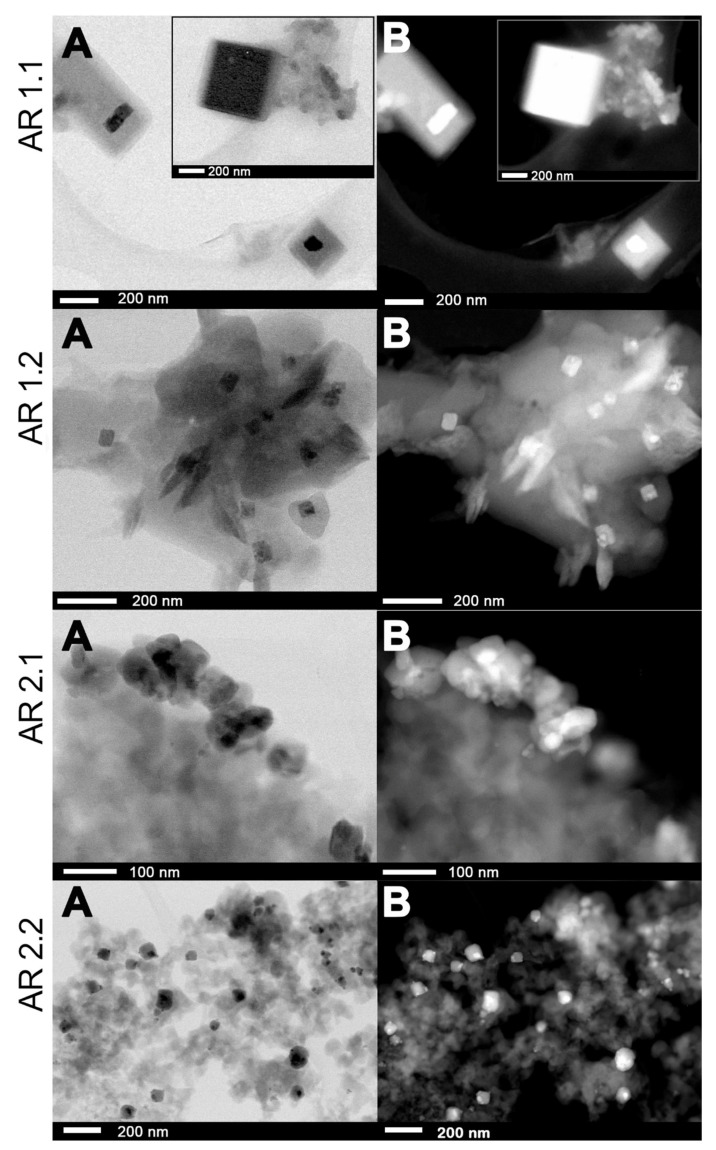
Comparison of the (**A**) bright and (**B**) dark field TEM image nanocomposites for envelope structure and square Ag nanoparticles.

**Figure 7 nanomaterials-14-00337-f007:**
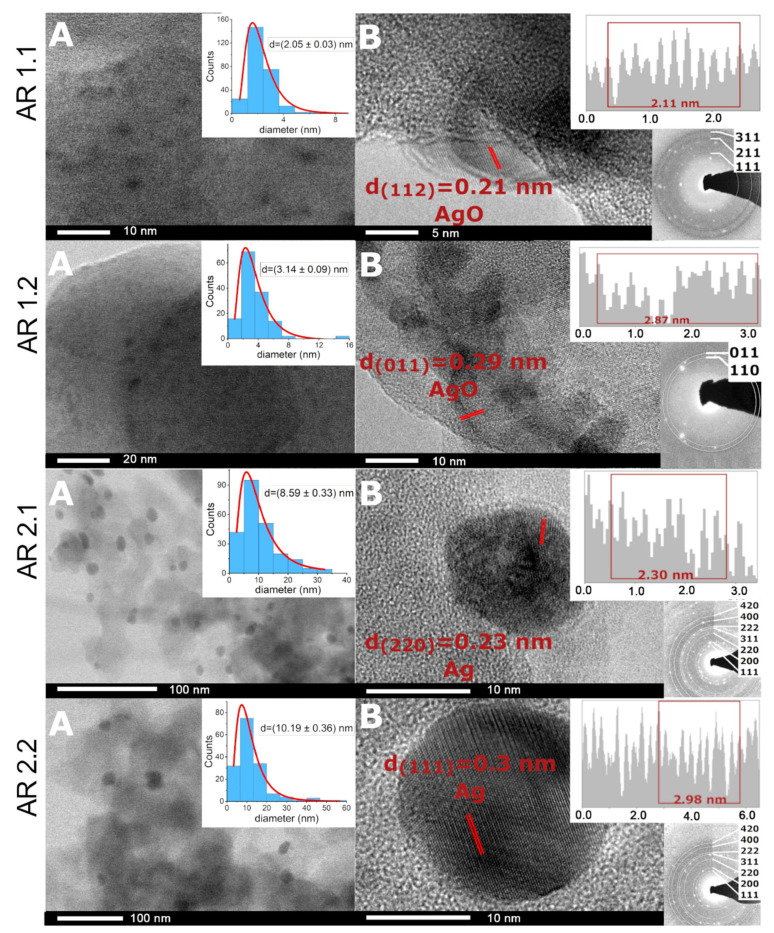
(**A**,**B**) HRTEM micro images for nanocomposites with the interplanar d-spacing plot with SAED pattern along the silver core and the particle size distribution of Ag nanoparticles. The data on histograms were fitted with a logN function.

**Figure 8 nanomaterials-14-00337-f008:**
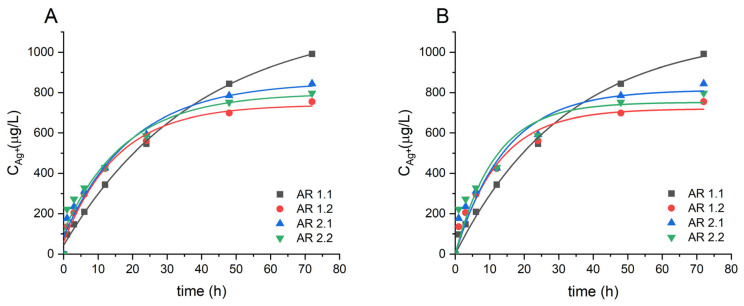
Cumulative release profiles of silver ions from all silver–silica nanocomposite samples were fitted using (**A**) an exponential model and (**B**) a pseudo-first-order model.

**Figure 9 nanomaterials-14-00337-f009:**
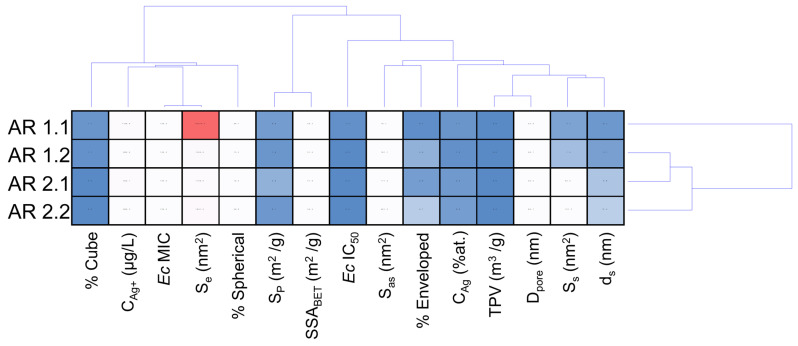
Cluster disposal of physicochemical variables of engineered nanomaterials and toxicological parameters for *Escherichia coli*. Individual abbreviations correspond to the content of ion released from the silver ion concentration released from the carrier (C_Ag+_ (μg/L)), silver concentration (C_Ag_ (at.%)), average diameter of spherical nanoparticles (d_s_), the surface area of spherical particles (S_s_), aspherical particles (S_as_), enveloped structures (S_e_), specific surface area (SSA_BET_), micropore volume (V_p_), micropore surface (S_p_), pore diameter (D_pore_), and total pore volume (TVP).

**Figure 10 nanomaterials-14-00337-f010:**
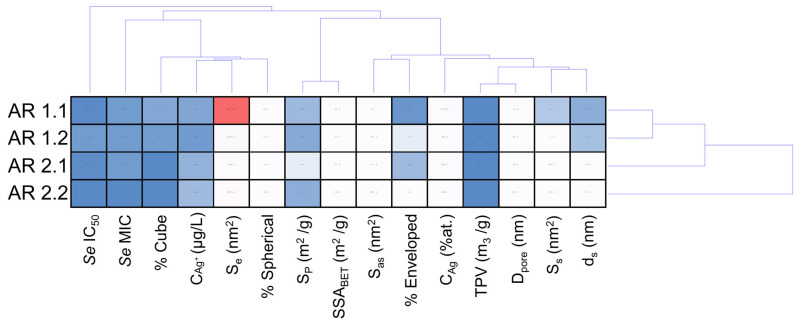
Cluster disposal of physicochemical variables of engineered nanomaterials and toxicological parameters for *Staphylococcus epidermidis*. Individual abbreviations correspond to the content of ion released from the silver ion concentration released from the carrier (C_Ag+_ (μg/L)); silver ion concentration (C_Ag_ (at.%)); average diameter of spherical nanoparticles (d_s_); the surface area of spherical (S_s_); aspherical particles (S_as_); enveloped structures (S_e_); specific surface area (SSA_BET_); micropore volume (V_p_); micropore surface (S_p_); pore diameter (D_pore_); and total pore volume (TVP).

**Table 1 nanomaterials-14-00337-t001:** The samples’ acronyms were prepared at variable synthesis conditions.

Samples’ Acronyms	AR 1.1	AR 1.2	AR 2.1	AR 2.2
Power (W)	800		150	
Time (s)	60	150	60	150

**Table 2 nanomaterials-14-00337-t002:** An averaged hydrodynamic silica particle diameter with standard deviation was obtained for the individual microwave-treated samples at different sonification times (mean ± SD).

Sonication Time (min)	AR 1.1	AR 1.2	AR 2.1	AR 2.2
	d ± Δd (nm)	d ± Δd (nm)	d ± Δd (nm)	d ± Δd (nm)
5	631 ± 14	555 ± 23	570 ± 30	507 ± 16
10	519 ± 6	458 ± 14	481 ± 20	475 ± 15
15	430 ± 5	401 ± 6	460 ± 20	467 ± 15
20	426 ± 14	376 ± 14	444 ± 6	387 ± 10
25	360 ± 9	365 ± 7	358 ± 13	392 ± 7

**Table 3 nanomaterials-14-00337-t003:** Nitrogen sorption parameters for Ag-SiO_2_ nanocomposite prepared at variable microwave power (P = 150 and 800 W) and time (t = 60 and 150 s). SSA_BET_—a specific surface area with standard deviation; V_p_—micropore volume; S_p_—micropore surface; D_pore_—pore diameter; TVP—total pore volume and porosity *: t—plot micropore volume. **: t—plot micropore area.

Samples’ Acronyms	SSA_BET_ (m^2^/g)	V_P_ (mL/g) *	S_P_ (m^2^/g) **	D_pore_ (nm)	TPV (cm^3^/g)	Porosity (%)
AR 1.1	32.28 ± 0.45	0.000545	2.76	23.17	0.186952	30.97
AR 1.2	31.53 ± 0.17	0.000379	1.81	28.69	0.226225	35.19
AR 2.1	37.79 ± 0.21	0.002323	5.78	29.28	0.276651	39.90
AR 2.2	32.46 ± 0.30	0.000386	2.11	69.73	0.565860	57.59

**Table 4 nanomaterials-14-00337-t004:** Sorption potential of silver ions of microwave-treated silicas. C_Ag_—silver ion concentration sorbed by the silicas; t_Ag_—time of the silver ion sorption.

Samples’ Acronyms	AR 1.1	AR 1.2	AR 2.1	AR 2.2
C_Ag_ (ppm)	3.94 ± 0.29	5.32 ± 0.30	1.54 ± 0.25	4.31 ± 0.42
t_Ag_ (min)	75.53 ± 9.82	72.20 ± 7.62	45.37 ± 21.68	43.90 ± 13.10

**Table 5 nanomaterials-14-00337-t005:** Content of individual phases in the tested samples and crystallinity index (mean ± SD).

Phase (Space Group)	AR 1.1	AR 1.2	AR 2.1	AR 2.2
	(%)	(%)	(%)	(%)
Amorphous SiO_2_	97.4 ± 0.1	96.9 ± 0.1	96.9 ± 0.1	96.4 ± 0.1
Ag_2_CO_3_ (P3_1_c)	0.4 ± 0.1	0.5 ± 0.1	0.4 ± 0.1	1.8 ± 0.1
Ag_2_CO_3_ (P2_1_/m)	1.5 ± 0.1	2.1 ± 0.1	2.0 ± 0.1	1.2 ± 0.1
Ag	0.7 ± 0.1	0.5 ± 0.1	0.7 ± 0.1	0.6 ± 0.1
Crystallinity index (%)	2.6 ± 0.2	3.1 ± 0.2	3.1 ± 0.1	3.6 ± 0.1

**Table 6 nanomaterials-14-00337-t006:** Lattice parameters of the silver phases were determined from the Rietveld refinement.

Phase (Space Group)	Ag_2_CO_3_ (P2_1_/m)				Ag_2_CO_3_ (P3_1_c)		Ag (Fm-3m)
Lattice parameters	a_0_ (Å)	b_0_ (Å)	c_0_ (Å)	β (°)	a_0_ (Å)	c_0_ (Å)	a_0_ (Å)
AR 1.1	3.256(4)	9.547(6)	4.854(7)	92.04(7)	9.191(1)	6.390(7)	4.089(1)
AR 1.2	3.254(1)	9.542(4)	4.851(7)	92.03(1)	9.185(2)	6.394(8)	4.089(2)
AR 2.1	3.254(3)	9.541(9)	4.852(8)	92.01(1)	9.185(7)	6.395(2)	4.088(9)
AR 2.1	3.256(7)	9.547(6)	4.855(1)	92.03(4)	9.192(6)	6.396(8)	4.089(4)

**Table 7 nanomaterials-14-00337-t007:** Crystallite size and strain of the silver phases.

Phase (Space Group)		AR 1.1		AR 1.2		AR 2.1		AR 2.2	
		Crystallite Size (Å)	Strain (%)	Crystallite Size (Å)	Strain (%)	Crystallite Size (Å)	Strain (%)	Crystallite Size (Å)	Strain (%)
Ag_2_CO_3_ (P2_1_/m)	[020]	1148	0.342	1113	0.349	1246	0.324	1109	0.350
	[110]	829	0.383	599	0.482	690	0.435	636	0.462
	[101]	539	0.319	482	0.347	512	0.331	488	0.343
Ag_2_CO_3_ (P3_1_c)	[110]	1348	0.297	866	0.397	1074	0.344	971	0.367
	[300]	674	0.268	693	0.262	543	0.313	710	0.258
Ag (Fm-3m)	[111]	285	0.465	266	0.491	263	0.496	180	0.687

**Table 8 nanomaterials-14-00337-t008:** Average atomic concentration for Ag-SiO_2_ nanocomposites estimated using TEM-EDS and SEM-EDS with statistical analysis (mean ± SD. *n* = 3: SEM. TEM).

Samples’ Acronyms	TEM-EDS			SEM-EDS		
	O (at.%)	Si (at.%)	Ag (at.%)	O (at.%)	Si (at.%)	Ag (at.%)
AR 1.1	83.0 ± 8.1	14.8 ± 9.2	2.2 ±1.8	76.2 ± 6.6	22.1 ± 7.0	1.7 ± 1.3
AR 1.2	81.3 ± 2.3	11.0 ± 5.3	7.8 ± 2.6	77.6 ± 0.5	21.5 ± 0.5	1.0 ± 0.1
AR 2.1	79.0 ± 3.5	20.3 ± 3.3	0. 7 ± 0.5	79.7 ± 3.9	18.0 ± 4.4	2.4 ± 1.4
AR 2.2	82.1 ± 1.6	16.8 ± 1.3	1.1 ± 0.4	80.2 ± 1.1	17.0 ± 2.6	2.8 ± 1.5

**Table 9 nanomaterials-14-00337-t009:** Comparison of the diameter of spherical Ag nanoparticles (d_s_) with statistical error estimated based on the logN function fit concerning the percentage of sphericity Ag nanoparticles obtained for Ag-SiO_2_ nanocomposites prepared at different microwave power (P = 150 and 800 W) and time (t = 60 and 150 s). The surface area of Ag nanoparticles obtained from the logN fit spherical (S_s_) and aspherical shape objects (S_as_). * The shape is illustrated in Figure 5.

Samples’ Acronyms	d_s_ (nm)	S_s_ (nm^2^)	S_as_ (nm^2^)	%Spherical	%Enveloped *	%Cube *
AR 1.1	2.05 ± 0.03	3.66 ± 0.10	74.14 ± 2.19	94.70	0.71	1.68
AR 1.2	3.14 ± 0.09	7.47 ± 0.26	161.66 ± 3.91	73.30	5.76	1.05
AR 2.1	8.59 ± 0.33	59.45 ± 5.23	171.18 ± 20.03	71.96	2.8	0
AR 2.2	10.19 ± 0.36	107.33 ± 7.83	227.5 ± 8.09	58.80	9.74	0

**Table 10 nanomaterials-14-00337-t010:** The average value of diagonal, area square, or rectangular envelopes surrounding the Ag nanoparticles spread in Ag-SiO_2_ nanocomposites. * The median of areas of envelopes.

Samples’ Acronyms	Diagonal of the Envelope (nm)	Envelope Area (nm^2^) *
AR 1.1	415.84 ± 158.25	60,213.97 ± 26,288.26
AR 1.2	41.88 ± 8.46	1325.23 ± 311.13
AR 2.1	49.64 ± 4.23	1109.92 ± 470.14
AR 2.2	59.05 ± 18.43	1841.37 ± 1338.04

**Table 11 nanomaterials-14-00337-t011:** Comparison of the exponential and pseudo-first order kinetic parameters release Ag ions for Ag-SiO_2_ nanocomposites. C_Ag+_—silver ion concentration released from the carrier (μg/L); t—time of ion release (hours); q_e_—the number of ions after equilibrium of adsorption (µg/L); q(t)—the number of desorbed ions at time t (µg/L); k_1_—pseudo-first rate constant (1/h).

	Exponential Model			Pseudo-First Order			
Samples’ Acronyms	C_Ag_ (µg/L)	t (h)	R^2^	q_e_ (µg/L)	k_1_ (1/h)	k_1_ (h)	R^2^
AR 1.1	1124.75 ± 66.03	38.99 ± 5.20	0.997	1097.50 ± 69.55	0.03 ± 0.004	33.3 ± 4.4	0.991
AR 1.2	680.22 ± 37.75	15.58 ± 2.56	0.986	719.68 ± 36.59	0.08 ± 0.01	12.5 ± 1.6	0.973
AR 2.1	852.74 ± 56.63	19.83 ± 4.46	0.978	813.50 ± 57.48	0.07 ± 0.01	14.3 ± 2.1	0.952
AR 2.2	684.25 ± 69.86	17.76 ± 5.41	0.957	752.22 ± 64.08	0.09 ± 0.02	11.1 ± 2.5	0.911

**Table 12 nanomaterials-14-00337-t012:** The MIC and IC_50_ values for *Escherichia coli* ATCC^®^ 25922^TM^ and *Staphylococcus epidermidis* ATCC^®^ 12228^TM^ treated with nanostructures (mg/L; *n* = 3; mean ± SD).

Samples’ Acronyms	*E. coli* ATCC^®^ 25922^TM^		*S. epidermidis* ATCC^®^ 12228^TM^	
	MIC	IC_50_	MIC	IC_50_
AR 1.1	180	13.34	120	9.88
AR 1.2	150	61.76	120	30.27
AR 2.1	150	47.28	120	12.33
AR 2.2	150	38.68	80	8.72

## Data Availability

Data are stored in the cloud and stuck as a backup.

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
