# Peer review of "Multifaceted Assessment of Porous Silica Nanocomposites: Unraveling Physical, Structural, and Biological Transformations Induced by Microwave Field Modification"

_nanomaterials, 2024, doi:10.3390/nano14040337_

Round 1
Reviewer 1 Report
Comments and Suggestions for Authors
Manuscript ID: nanomaterials-2824914
Title: Multifaceted Assessment of Porous Silica Nanocomposites: Unraveling
Physical, Structural, and Biological Transformations Induced by Microwave
Field Modification
Authors: Aleksandra Strach *, Mateusz Dulski *, Daniel Wasilkowski, Krzysztof
Matus, Karolina Dudek, Jacek Podwórny, Patrycja Rawicka, Vladlens Grebnevs,
Natalia Waloszczyk, Anna Nowak, Paulina Poloczek, Sylwia Golba
The manuscript reports the fabrication of silver-loaded mesoporous silica particles. Four types of particles formed at different powers and durations of microwave irradiation were studied. The adsorption of silver on each type of particles was carried out by varying the duration of ultrasonic exposure from 5 to 25 minutes. The sample preparation method influenced the particle size, pore diameter, amount of adsorbed silver, the rate of release of silver into the aqueous solution, and the final antimicrobial activity of the samples. The problem addressed in the article is both interesting and relevant to the current state of research in the field. The experimental methods used can produce reliable and valuable results.
Nevertheless, the version of the article submitted for review lacks the opportunity to assess the thoughtfulness of the chosen sample preparation strategy, and it does not provide a means for evaluating the merits of each investigated sample. The text is intricate, causing confusion, and lacks a smooth logical flow.
The text needs to be shortened.
At the beginning of each section, it is necessary to explain which property will be studied. At the end of the section, give a short description of the results obtained.
Move all secondary results to the Supporting Information.
In the concluding section, succinctly outline the specific results achieved and offer a summary of the optimal strategy employed to acquire this type of materials.
Minor improvements to be made.
The meaning of many sentences is challenging to comprehend.
“It has been reported that accumulations of silver and inflammation in the lungs of Sprague-Dawley and Brown-Norway rats [14].”
“Indeed, toxicological studies in organisms ranging from zebrafish to freshwater worms impair osmoregulation [15].”
“In a study conducted on Daphnia magna, the toxicity of small-diameter Ag NPs was shown to be higher [23].”
“According to different fabrication routes, green-synthesized Ag NPs generally exhibit lower cytotoxicity than their chemically synthesized counterparts [19].”
“Micro- (<2 nm) and mesoporous silica (2–50 nm) … The primary objective revolves around fabricating nanocomposites adept at absorbing silver compounds in the form of stable, bioactive Ag2O/Ag2CO3 heterojunctions within the microporous silica. … Diameter of silica nanoparticles (dSiO2 = 15–20 nm, pore size = 2-6 nm”
Explain the choice of duration of microwave irradiation, rename the samples so that their names are associated with the method of preparation. What has been done previously in this area? Provide a concise summary of the structural properties for each of the samples in the conclusion.
Revise the presentation of experimental data uncertainty throughout to ensure clarity and appropriateness. 631.20 +- 13.54 is 631 +- 14
Comments on the Quality of English LanguageSee the review report.
Author Response
The manuscript reports the fabrication of silver-loaded mesoporous silica particles. Four types of particles formed at different powers and durations of microwave irradiation were studied. The adsorption of silver on each type of particles was carried out by varying the duration of ultrasonic exposure from 5 to 25 minutes. The sample preparation method influenced the particle size, pore diameter, amount of adsorbed silver, the rate of release of silver into the aqueous solution, and the final antimicrobial activity of the samples. The problem addressed in the article is both interesting and relevant to the current state of research in the field. The experimental methods used can produce reliable and valuable results.
Nevertheless, the version of the article submitted for review lacks the opportunity to assess the thoughtfulness of the chosen sample preparation strategy, and it does not provide a means for evaluating the merits of each investigated sample. The text is intricate, causing confusion, and lacks a smooth logical flow.
→ Thank you for the comment. We tried to modify the text to make it more transparent without unnecessary data and information. As you can find, practically all the text is yellow-marked illustrating corrected fragments. In our opinion, the version in the present form seems to be better, and we hope that, in your opinion, the text has been significantly improved.
The text needs to be shortened.
→ According to Your suggestion, some fragments were shortened and written more clearly, in our opinion.
At the beginning of each section, it is necessary to explain which property will be studied. At the end of the section, give a short description of the results obtained.
→ According to Your suggestion, appropriate sentences were added to clarify the text fragment in our opinion.
Move all secondary results to the Supporting Information.
→ According to Your suggestion, some information was removed or transferred to the supplementary data.
In the concluding section, succinctly outline the specific results achieved and offer a summary of the optimal strategy employed to acquire this type of materials.
→ According to Your suggestion, the conclusions were rewritten, and we hope that the present version is more clarified and summarizes data better than previously.
Minor improvements to be made.
The meaning of many sentences is challenging to comprehend.
“It has been reported that accumulations of silver and inflammation in the lungs of Sprague-Dawley and Brown-Norway rats [14].”
“Indeed, toxicological studies in organisms ranging from zebrafish to freshwater worms impair osmoregulation [15].”
“In a study conducted on Daphnia magna, the toxicity of small-diameter Ag NPs was shown to be higher [23].”
“According to different fabrication routes, green-synthesized Ag NPs generally exhibit lower cytotoxicity than their chemically synthesized counterparts [19].”
“Micro- (<2 nm) and mesoporous silica (2–50 nm) … The primary objective revolves around fabricating nanocomposites adept at absorbing silver compounds in the form of stable, bioactive Ag2O/Ag2CO3 heterojunctions within the microporous silica. … Diameter of silica nanoparticles (dSiO2 = 15–20 nm, pore size = 2-6 nm”
→ Thank you for your comments. Many of those sentences were rewritten, and the text was subjected to specialized proofreading.
Explain the choice of duration of microwave irradiation, rename the samples so that their names are associated with the method of preparation. What has been done previously in this area? Provide a concise summary of the structural properties for each of the samples in the conclusion.
→ According to Your suggestion, the Introduction and Conclusions were rewritten. The Introduction added new fragments explaining the problem of the microwave treatment, while the whole conclusion was modified to sum up all the data discussed in the paper. We hope that in the present version, all the information is more clarified and summarizes data better than previously.
→ We decided to perform studies at such conditions due to a few factors: i) those conditions are part of our step-by-step studies, and in future articles, we will discuss more in detail the effect of microwave-assisted sintering at other times and powers; ii) our idea was to use highly two different fields and times to see how a small field and an increase in time changes our material and affect its effect on microorganisms in contrast to Díaz de Greñu et al. (https://doi.org/10.3390/nano10061092) in which reported impact of microwave field on silica exposed to 450W for one h. Additional fragments were also added to the methodology to explain our idea better.
Revise the presentation of experimental data uncertainty throughout to ensure clarity and appropriateness. 631.20 +- 13.54 is 631 +- 14
→ Thank you for the comment. The data has been corrected.

Reviewer 2 Report
Comments and Suggestions for Authors
Dear authors,
Please explain, for the cumulative release profiles of silver ions, how did you calculate the cummulative concentration of silver ions (the release at Tx plus all the release from To to Tx).
How can be possible explain the influence of microwave on the resulted particle spericity It will be useful to add a discussion and possible explanation concening the solvent polarity and microwave action.
What about the particle rugosity? The post probably the microvafe action produce more rogh surfaces (as other literature studies proved by now). Could you please give the FHH parameters resulted from your porosimetry data or if possible, from scattering measurements this information could me easily provide.
Is it possible to calculate the total energy transferred to your system and subsequently to calculate the energy in Jouly/mL that you applied to your sysnthesis system?
Please underline in the conclusions the overall advantages (like increase adsorption capacity, more uniform distribution, with underlining why this is an advantage) of using these obtained silica particle in adsorption, compared wit simple materials for what ultasounds and/or microvawes irradiation was not necessary.
Author Response
Please explain, for the cumulative release profiles of silver ions, how did you calculate the cummulative concentration of silver ions (the release at Tx plus all the release from To to Tx).
→ Thank you for the comment. We discussed such a question with co-authors; unfortunately, the text has introduced an incorrect term ("cumulative release profiles"), which introduced other problems about which you asked. Hence, we modified the text to be more clarified, and we hope that we have done this information appropriately. All symbols are (we hope) explained at appropiate equations.
How can be possible explain the influence of microwave on the resulted particle spericity It will be useful to add a discussion and possible explanation concening the solvent polarity and microwave action.
→ Thank you for the comment. We tried to modify the text according to your suggestion, and therefore, we have added new information in the Introduction. Moreover, we have tried to discuss the problem of sphericity in the text more precisely. Therefore, many fragments were yellow-marked. In our experiment, we used the polar solvent (water) with a low dissipation factor (εv/ε0) and low capable of converting electromagnetic energy into heat according to Kappe C. 2004 (10.1002/anie.200400655) or Nithya T. et al. 2019 (10.1016/B978-0-12-817813-3.00004-3). We hope that we successfully corrected the fragments.
What about the particle rugosity? The post probably the microvafe action produce more rogh surfaces (as other literature studies proved by now).
→ Thank you for the comment. We tried to modify the text according to your suggestion, and therefore, we have added new information in the Introduction.
Could you please give the FHH parameters resulted from your porosimetry data or if possible, from scattering measurements this information could me easily provide.
→ Thank you for the comment. We want to do that, but unfortunatelly, our system is a more basic version and gives only information in the range of 0-0.3. Additionally, we can't gather desorption isotherm and estimate the volume adsorbed at the relative equilibrium pressure P/Po, and Vm of the monolayer, which are essential according to the FHH equation (10.1016/j.jngse.2014.10.018).
Is it possible to calculate the total energy transferred to your system and subsequently to calculate the energy in Jouly/mL that you applied to your sysnthesis system?
→ Thank you for a valuable comment. Taking into account our present data, unfortunately not. However, it is a fascinating idea for further investigations.
Please underline in the conclusions the overall advantages (like increase adsorption capacity, more uniform distribution, with underlining why this is an advantage) of using these obtained silica particle in adsorption, compared wit simple materials for what ultasounds and/or microvawes irradiation was not necessary.
→ Thank you for a valuable comment. According to your suggestion and the comment of the second Reviewer, the Conclusion fragment was rewritten entirely. We hope that in the newest version, the conclusions summarized data much better.

Round 2
Reviewer 1 Report
Comments and Suggestions for Authors
I have no more comments that would necessitate another review cycle.